# Vector Graveyards: A Self-Pruning Knowledge Graph as Shared Memory for a Human-LLM Security Agent

Dipankar Sarkar[1]

[1]*Independent Researcher*

## Abstract

Autonomous LLM agents lose track of what they have already tried: across long-running loops they re-explore dead ends, forget *why* a path failed, and leave the human operator unable to audit their reasoning. We report on *Hack Monty*, a deployed security-assessment agent, and on its redesign around a lightweight SQLite *knowledge graph* that serves as shared memory between the LLM, its curating sub-agents, and a human analyst. The system rests on two design choices. First, knowledge is *derived, not asserted*: a SQL view computes each attack vector's status (`untried`/`active`/`promising`/`dead`) from attempt history, and the agent's governing rule is "never probe what the graph already knows." Second, every contribution is *provenance-bearing*: each run must supply a *reason*, and findings carry references to the web and CVE sources that motivated them. Reconstructing the graph from 679 logged attempts, we find that the derivation rule would retire 13 vectors as "dead," and that 566 attempts (83%) were avoidable re-probes of vectors already past that threshold. This quantifies the redundancy the graph is designed to suppress. We present the work as a case study of Human-LLM-KG collaboration, and we are explicit about where a relational graph falls short of a semantic one.

## Keywords

human–LLM–KG collaboration, agent memory, knowledge graph, derived knowledge, autonomous agents, LLM tool use, security assessment

## 1. Introduction

Large language model (LLM) agents are increasingly deployed in long-running, open-ended loops: they propose an action, observe a result, and decide what to do next, for hundreds or thousands of iterations. A recurring failure mode of such loops is *amnesia*. Within a single context window the agent can reason about recent steps, but as the loop continues it forgets which approaches it has already exhausted, re-derives conclusions it reached an hour earlier, and, for any human supervising it, produces no durable, queryable record of *why* it did what it did. Vector stores and retrieval-augmented generation [1] mitigate the recall problem for unstructured text, but they do not naturally express the kind of negative, relational knowledge an exploratory agent most needs: *this class of attempt is dead; that one is still open; this finding came from that source.*

This paper is an experience report on giving such an agent a structured, shared memory in the form of a small *knowledge graph* (KG), and on what that changed. Our setting is *Hack Monty*, an autonomous security-assessment agent built to attack Pydantic's "Monty" sandbox, a Rust reimplementation of a Python interpreter, as part of a public $10,000 bounty [14, 13]. The agent submits Python programs to a sandbox honeypot, scores the outcome on a fixed 0–5 scale, and iterates. Over four design generations the full campaign exceeded 750 exploit attempts; 679 were preserved in the durable logs and form the basis of the analysis in this paper. The version we report on here (V4) replaced an ad-hoc collection of note files and a multi-armed-bandit selector [17] with a SQLite-backed KG that all participants (the LLM orchestrator, three curating sub-agents, and the human analyst) read from and write to through a small set of tools exposed over the Model Context Protocol (MCP) [12]. We position the work as an In Use and Experience report rather than a controlled evaluation: its contribution is a design, a deployment, and the lessons drawn from them.

*UKG 2026: 2nd International Workshop on Users and Knowledge Graphs, September 15, 2026, Ghent, Belgium (co-located with SEMANTiCS 2026)*

✉ me@dipankar.name (D. Sarkar)

🌐 https://dipankar.name/ (D. Sarkar)

We call this artifact a knowledge graph while being precise about what it is: a *KG-shaped, relationally materialised interaction graph* used as a user-facing collaboration substrate, rather than a semantic (RDF/OWL) graph in the classical sense [5]. Its nodes are attempts, derived attack vectors, and discoveries; its edges are the relations between them and the references that ground discoveries in external sources. The relational design is a deliberate minimal KG substrate, chosen so that the most-used knowledge can be a continuously maintained *derived view* (§4); a future semantic version would make attack techniques, provenance, and confidence first-class (§8). Even at this weight, two design decisions make it an effective collaboration substrate, and both speak directly to the themes of the UKG workshop: how users, human and machine, query, explore, and contribute to a KG.

**Contribution 1: knowledge is derived, not asserted.** The single most-used piece of knowledge in the system, whether an attack vector is worth trying, is never written down by anyone. It is computed on read by a SQL view that classifies each vector as `untried`, `active`, `promising`, or `dead` from the history of attempts against it. The orchestrator's governing instruction is direct: *never probe what the dashboard already knows.* The graph therefore prunes the search space rather than only storing it; we call the set of auto-retired vectors the *vector graveyard.*

**Contribution 2: every contribution is provenance-bearing.** The KG is write-guarded in a way that suits an LLM author. Every code execution *must* carry a natural-language *reason*, from which the attack vector is extracted and under which the attempt is filed; discoveries carry a `refs` array of the web pages and CVE advisories that motivated them. The graph is therefore simultaneously the agent's working memory and a human-auditable trail.

**Findings.** Reconstructing the graph from the 679 attempt records preserved in the repository, we find that the derivation rule retires 13 of 42 vectors as `dead`. Counting precisely, **566 attempts (83%) were avoidable re-probes**: attempts against a vector that had already crossed the rule's retirement threshold of two scoreless tries. This quantifies, on real deployment data, how much re-probing a "never-re-probe" KG is *designed* to suppress. We do not claim a measured speed-up, having run no A/B comparison (§7). The reconstruction also surfaces a limitation: LLM-authored vector keys are noisy, and a majority of the graph's distinct nodes are malformed because the model drifted from the expected key format, though these nodes carry only a small fraction of the attempts. We draw both positive and negative lessons for anyone building Human-LLM-KG collaboration loops.

## 2. Background and Related Work

**Memory for LLM agents.** Agent frameworks such as ReAct [2] and Reflexion [3] interleave reasoning and acting, and the latter explicitly keeps a memory of past failures to avoid repeating them, but that memory lives in the context window or as free text. Generative Agents [4] introduced a structured memory stream with reflection and retrieval. The dominant production pattern remains retrieval-augmented generation over a vector store [1]: effective for "what did I read about X" but a poor fit for relational, negative, and derived facts such as "vector D failed twice and is now dead." Our work uses a structured graph, rather than an embedding index, as the agent's primary memory. The difference is clearest for *negative* knowledge: an embedding index answers "what is similar to this" but has no native notion of "what has been ruled out," which for an exploratory agent is the knowledge that prevents wasted work. A graph whose schema makes ruled-out a queryable status closes that gap.

Table 1 situates our choice against the two dominant memory substrates for LLM agents along the dimensions an exploratory agent cares about.

**Knowledge graphs and their users.** Knowledge graphs organise information as entities and typed relations and have a mature ecosystem of query, validation, and visualization tooling [5]. In practice, the hardest problems in deploying KGs are as much about the people who build, trust, and maintain

**Table 1**

Memory substrates for LLM agents, on the dimensions an exploratory agent cares about. "Negative knowledge" is the ability to represent and query *what has been ruled out*.

| Property | Context window | Vector store / RAG | Derived KG (ours) |
|---|---|---|---|
| Survives long loops | not durably | yes | yes |
| Negative knowledge | implicit | weak | first-class |
| Relational queries | no | non-native[†] | yes |
| Provenance / audit | no | partial | yes (refs + reason) |
| Staleness of status | n/a | manual | none (derived) |
| Setup weight | none | medium | low (one file) |

[†]Hybrid systems (metadata-filtered RAG, GraphRAG [9]) add structure, but relational/path queries are not native to an embedding index.

them as about the formalism, a point made repeatedly in industry experience reports [7, 8]. The UKG workshop's premise is that the *user*'s interaction with a KG, how they query it, trust it, and contribute to it, is as important as the graph itself, and that LLMs are becoming a new class of KG user mediating between humans and graphs. Our system is one such instance: a principal "user" is an LLM that both reads the graph (to decide what to do) and writes to it (to record what happened), under guardrails designed for a non-human author. The broader programme of unifying LLMs and KGs, using graphs to ground generation and using models to populate graphs, is surveyed by Pan et al. [6]; our contribution is a narrow point in that space where the graph's job is to encode *what not to do.*

**Autonomous security agents.**  LLM-driven penetration testing has advanced quickly, from Pentest-GPT's guided task trees [10] to commercial autonomous offensive agents such as XBOW [11]. These systems face exactly the memory problem we target: a long campaign generates many probes whose results must inform later ones. Hack Monty began from Karpathy's *autoresearch* loop [16] and evolved toward the present KG-centric design. The target, Monty, is a Rust-based Python sandbox; its first bounty round was won via a use-after-free in `list.sort(key=func)` [15], which frames the kinds of attack vectors our agent explores.

**The tool boundary.**  The agent reaches the graph and the sandbox through the Model Context Protocol [12], which exposes typed tools to an LLM. MCP matters here because it is the surface at which we enforce the write guards (the mandatory *reason*) and expose read-only query access. The protocol is the KG's user interface for its machine user.

## 3. The Hack Monty System

Hack Monty is an autonomous loop that attacks the Monty sandbox honeypot at `hackmonty.com` by POSTing Python programs to it and inspecting the result [14]. Figure 1 shows the V4 architecture. An LLM agent harness drives the loop; a set of role-specialised *skills* (each a system prompt plus an allow-list of tools) lets a top-level *orchestrator* spawn focused sub-agents. All side effects (running code, scoring, reading and writing the knowledge graph, web research) pass through a single MCP server (`hackmonty_mcp_server.py`), which owns the SQLite database and the sandbox client.

**The loop.**  Each iteration the orchestrator (i) reads the KG dashboard to learn what is dead and what is promising, (ii) optionally researches a technique on the web and records it, (iii) runs a candidate exploit with a stated reason, (iv) receives a 0–5 score, and (v) either drills into a promising result or moves on. The scoring rubric is fixed and rule-based to prevent the model from rewarding itself: 0 "standard error / expected sandbox behaviour," 1 "crash/panic," 2 "interesting / unexpected snapshot," 3 "host info leak," 4 "filesystem read," 5 "secret found." Because scores are computed by `evaluate.py` from the

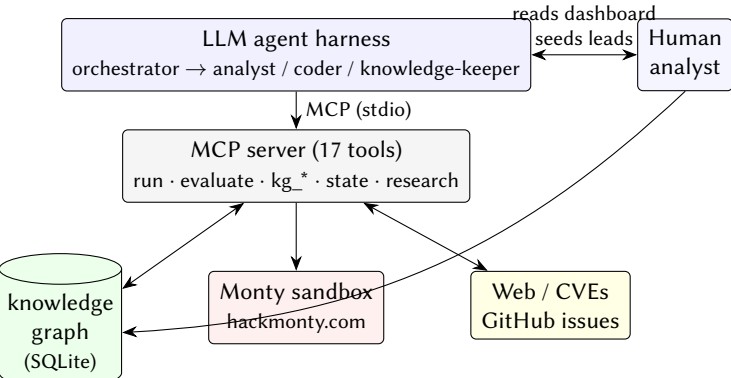

**Figure 1:** V4 architecture. The MCP server is the single boundary; the knowledge graph is the shared memory read and written by the LLM agents and inspected by the human analyst.

**Table 2**

The fixed 0–5 scoring rubric. Scores are computed from the sandbox response by `evaluate.py`, not chosen by the LLM, so the graph's status derivation rests on trustworthy values.

| Score | Label | Trigger |
|-------|-------|---------|
| 0 | Nothing | Standard Python/sandbox error or expected behaviour |
| 1 | Crash/Panic | Segfault, stack overflow, or Rust panic |
| 2 | Interesting | Unexpected snapshot kind or internal error |
| 3 | Host Info | Rust traceback or host path leaked |
| 4 | Filesystem | Non-public file content reaches the output field |
| 5 | Secret Found | Target secret extracted (campaign success) |

sandbox response rather than by the LLM, the values written to the graph are trustworthy inputs to the derivation rule of §4. Table 2 gives the rubric in full.

**The sandbox boundary.**   Monty pauses execution at every external call through a *snapshot/resume* protocol: the program runs until it touches the outside world, the interpreter emits a typed *snapshot* (a name lookup, a function call, a coroutine, a container access), and the host resolves it. These snapshot kinds are themselves a signal, since an unexpected kind can indicate the program reached a code path the sandbox did not intend, so the agent records the `snapshot_kinds` of every run into the graph alongside the score (§6).

**Operating a long LLM loop.**   Two production lessons shaped V4 and are worth recording for an experience report. First, model calls occasionally hang; we added a 120 s hard timeout (via a signal alarm) so a stuck call surfaces as a retryable error rather than freezing the campaign. Second, the harness's automatic context *compaction* (rewriting the running history to fit the window) triggered a double-free crash in the native runtime when it rewrote tool-call history, so we disabled it and accept longer contexts instead. Both decisions push more of the agent's working state out of the volatile context window and into the durable knowledge graph, which is part of why a structured external memory mattered in practice and not just in principle.

**Governance through tool allow-lists.**   Each skill declares the exact set of tools it may call. This doubles as an access policy on the graph: the coder has no KG tools at all, the analyst may write discoveries but not run code, and only the orchestrator may execute (and thus auto-write attempts). In KG terms, the write surface is partitioned by role at the protocol boundary, a lightweight way to keep a

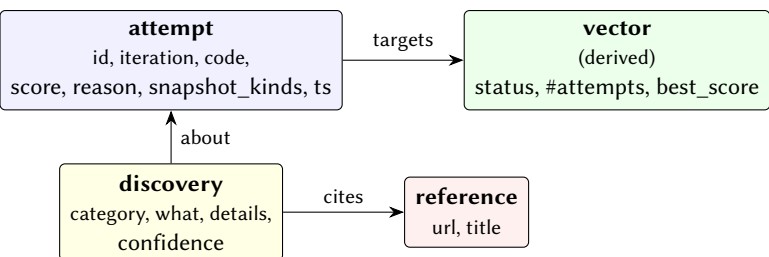

**Figure 2:** Entity model. *attempt* and *discovery* are stored relations; *vector* is a node materialised on read by the `vector_status` view; *reference* entries live inside a discovery's `refs` array and ground it in external sources.

multi-agent system's contributions to a shared graph well-scoped, without any in-graph permission model.

**Roles as KG users.** Four skills interact with the graph in distinct ways. The *orchestrator* is the primary reader-and-writer; the *analyst* researches techniques and writes *discoveries* with source references; the *coder* produces syntactically valid exploit code (and does not touch the graph); the *knowledge-keeper* is a periodic curator that reads recent activity and records derived discoveries (e.g., marking a vector class blocked). This division of producers, a curator, and a human analyst over one shared graph is the collaboration structure we examine in §5.

## 4. Knowledge Graph Design

The graph is deliberately small: two base relations and a set of views, all in a single SQLite file. Figure 2 sketches the entity model and Listing 1 gives the core schema.

Listing 1: Core schema: two base tables and the derived `vector_status` view (abridged from `hackmonty_mcp_server.py`).

```
CREATE TABLE attempts (
  id INTEGER PRIMARY KEY AUTOINCREMENT,
  iteration INTEGER NOT NULL,
  vector TEXT NOT NULL,              -- extracted from the run's reason
  code TEXT, score INTEGER,          -- 0..5, computed by evaluate.py
  reason TEXT NOT NULL,              -- WHY this was attempted (required)
  snapshot_kinds TEXT, timestamp TEXT DEFAULT (datetime('now')));

CREATE TABLE discoveries (
  id INTEGER PRIMARY KEY AUTOINCREMENT,
  category TEXT,                     -- blocked|available|technique|cve|finding
  what TEXT NOT NULL, details TEXT,
  refs TEXT DEFAULT '[]',            -- JSON [{url,title}] provenance
  confidence TEXT DEFAULT 'confirmed');

CREATE VIEW vector_status AS
  SELECT vector, COUNT(*) AS attempts, MAX(score) AS best_score,
    CASE WHEN MAX(score)=0 AND COUNT(*)>=2 THEN 'dead'
         WHEN MAX(score)>0                 THEN 'promising'
         WHEN COUNT(*)=0                   THEN 'untried'
         ELSE 'active' END AS status
  FROM attempts GROUP BY vector;
```

**Derived knowledge.** The central property of the design is that `vector_status` stores nothing. A vector becomes dead the moment its second scoreless attempt lands, becomes `promising` as soon as any attempt scores above zero, and is otherwise `active`. Because the classification is a view, it is always consistent with the underlying attempts, cannot go stale, and requires no curator to maintain.

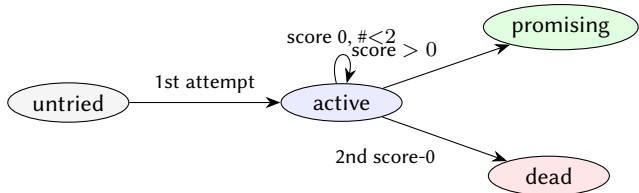

**Figure 3:** The `vector_status` state machine, evaluated on every read. "Dead" vectors form the *vector graveyard* the orchestrator is forbidden to revisit.

**Table 3**
Knowledge-graph tools exposed over MCP, and their primary user.

| Tool | Mode | Purpose / primary user |
|---|---|---|
| `hackmonty_run` | write | Run code; auto-records attempt with required *reason* (orchestrator). |
| `kg_discover` | write | Record a finding with source `refs` (analyst, knowledge-keeper). |
| `kg_dashboard` | read | Vector status + recent attempts + discoveries (orchestrator, human). |
| `kg_dead_vectors` | read | The graveyard: vectors never to retry (orchestrator). |
| `kg_recent` | read | Last $N$ attempts (orchestrator, knowledge-keeper). |
| `kg_query` | read | Arbitrary read-only SELECT (power users, ad-hoc analysis). |
| `kg_bootstrap` | admin | One-time import of legacy note files into the graph. |

The graph's most useful knowledge is a side effect of complete logging. Figure 3 shows the induced state machine.

**Provenance-bearing writes.** Writes are shaped for an LLM author. The execution tool's signature is effectively `hackmonty_run(code, reason)` with `reason` required; the server splits the reason to derive the `vector` key and records the attempt automatically, so the agent cannot run code that escapes the graph. Discoveries are written through `kg_discover(category, what, details, refs)`, where `refs` is a JSON array of {url, title} captured during web research, which makes every asserted finding traceable to a source. Table 3 lists the KG tools and the user role each primarily serves.

**Discoveries and their categories.** Where the `attempts` relation is the agent's episodic memory, the `discoveries` relation is its semantic memory: distilled, reusable facts about the target. Each discovery carries a `category` (one of `blocked`, a technique the sandbox refuses; `available`, a primitive that works; `technique`, a method worth trying; `cve`, a known vulnerability; or `finding`, a substantive result), a `confidence` level, and the `refs` provenance array. This is the part of the graph a human most directly reads and writes: an analyst seeds `technique` and `cve` discoveries to steer the agent, and the agent (via the analyst and knowledge-keeper roles) writes `blocked` discoveries back as it rules approaches out.

**Bootstrapping from legacy notes.** The graph did not start empty. Earlier system generations logged every attempt as a markdown file and every result as free text; `kg_bootstrap` performs a one-time migration, parsing those files into `attempts` and `discoveries` rows. This migration path is also what makes the present study possible: the same logic reconstructs the graph offline from the preserved corpus (§6). It carries a small lesson of its own. Because the durable record was human-readable text

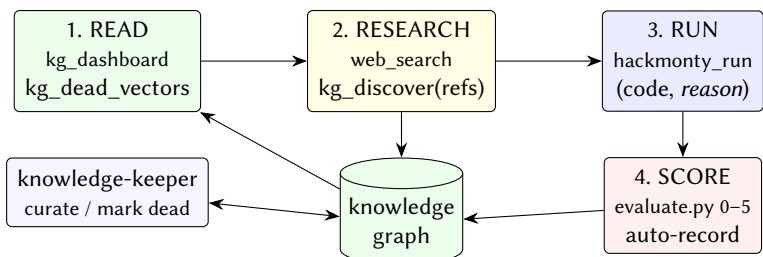

**Figure 4:** The collaboration loop. The graph is both the source the agent reads to plan (arrow up-left) and the sink every action writes to (arrows in). The knowledge-keeper curates asynchronously; the human reads the same dashboard.

rather than a database, the knowledge survived three architectural rewrites and could be re-imported into the new structure.

**What the agent actually reads.**   Listing 2 shows an abridged `kg_dashboard` payload, the single artifact the orchestrator consults at the top of every iteration. It combines the derived vector statuses, the most recent attempts, and the latest discoveries into one read, so a single tool call answers "what is dead, what is promising, what have we just learned, and where."

Listing 2: Abridged `kg_dashboard` payload. The `vectors` block is the derived state; `recent` and `discoveries` are episodic and semantic memory.

```
{ "vectors": [
    {"vector":"I","attempts":59,"best_score":1,"status":"promising"},
    {"vector":"F","attempts":108,"best_score":0,"status":"dead"}, ...],
  "recent": [
    {"id":679,"vector":"I","score":1,"reason":"I: name_lookup + open() ..."}, ...],
  "discoveries": [
    {"category":"blocked","what":"user-defined __eq__ unavailable",
     "refs":[{"url":"...","title":"Monty Round 2 notes"}]}, ...] }
```

**Read paths.**   Three read tools serve three granularities: `kg_dashboard` for the synoptic "what do we know" view the orchestrator consults each iteration; `kg_dead_vectors` for the explicit do-not-retry list; and `kg_query`, a guarded SELECT-only escape hatch that lets either the LLM or the human ask questions the fixed views do not answer (e.g., "which vectors produced `function_snapshot` kinds?"). The SELECT-only guard lets an LLM compose queries against the graph without any risk of writes.

## 5. Human-LLM-KG Collaboration

The workshop's central question is how users work *with* a knowledge graph, and increasingly how humans and LLMs do so together. Hack Monty is a small instance of this: a single graph mediates among an LLM producer, automated curators, and a human analyst. Figure 4 shows the loop.

Table 4 makes the collaboration concrete: four classes of user share one graph, each reading and writing a different slice, and each kept honest against a different failure mode. The KG is therefore a *users-and-KG* artifact rather than a private agent log, because distinct human and machine users meet on it.

**The LLM as a disciplined KG user.**   The orchestrator's standing instructions function as a usage policy for a capable but careless user. The first rule, *never probe what `kg_dashboard` already knows*, makes reading the graph a precondition for acting. The reason requirement makes every write self-documenting. The prohibition on re-trying dead vectors turns the graph from a passive log into an active constraint on the agent's behaviour. In KG terms, the agent is a user that is *required* to query

**Table 4**
The four users of the shared graph: what each reads, writes, the failure its discipline prevents, and the audit value its contributions leave behind.

| User | Reads | Writes | Failure prevented | Audit value |
|---|---|---|---|---|
| Orchestrator LLM | dashboard, dead vectors | attempts (auto) | re-probing dead vectors | reason trail |
| Analyst (sub-agent) | dashboard, discoveries | leads, refs | wrong search direction | provenance |
| Knowledge-keeper | recent attempts | blocked discoveries | stale / free-text memory | summaries |
| Human analyst | dashboard, logs | seeds, inspection | opaque agent loops | post-hoc audit |

before it contributes, and whose contributions are schema-shaped at the tool boundary rather than trusted to be well-formed.

**An iteration, concretely.**   Listing 3 shows one pass through the loop as a sequence of tool calls. The agent reads before it acts, states why it is acting, and the result is filed automatically; nothing the agent does escapes the graph, and a human reading the log afterwards can reconstruct the intent behind every probe.

Listing 3: A single orchestrator iteration, as KG-mediated tool calls. The *reason* both documents the attempt and supplies the vector key under which it is grouped.

```
kg_dashboard()                       # READ: F,C,A,...=dead; I=promising
kg_dead_vectors()                    # READ: do-not-retry list
web_search("monty name_lookup ...")  # RESEARCH
kg_discover(category="technique",    # WRITE w/ provenance
  what="name-lookup expr capture",
  refs=[{"url": "...", "title": "..."}])
hackmonty_run(                       # WRITE: auto-records attempt
  code="...",
  reason="I: combine name_lookup + open() to capture expr value")
# -> evaluate.py scores 1 (crash); vector I stays 'promising'
```

**Technique knowledge encoded for reuse.**   Beyond per-attempt records, the system encodes a small library of *combination patterns*: pairs of primitives that are more likely to break a sandbox together than alone (e.g. *name-lookup + file-read, sort + GC, exception + frame-walk, async + snapshot-chain*). These live in the orchestrator's instructions and surface in the graph as the reason text of attempts that try them, so a query for "which combinations have we tried and how did they score" is answerable directly. This motivates the typed *combines*-relation we propose making first-class in §8.

**Automated curation.**   The knowledge-keeper sub-agent embodies a second user role: maintenance. Spawned periodically, it reads `kg_recent` and the dashboard, identifies vectors that the derivation rule has retired but that have not yet been written up as `blocked` discoveries, and records those generalisations with references. This separates two kinds of knowledge cleanly: the *derived* status that needs no curator, and the *abstracted* discovery ("this whole class of attack is blocked because Monty lacks user-defined `__eq__`") that benefits from one.

**The human in the loop.**   The human analyst is a first-class user of the same graph. They consult `kg_dashboard` to see where effort is going, seed high-value leads, and read the final synthesis. A concrete example: knowing that the previous bounty round was won through a use-after-free in `list.sort(key=func)` [15], the analyst writes a `technique` discovery prioritising the sort-plus-GC class and a `cve` discovery citing the write-up; on its next read the orchestrator sees the lead at the

**Table 5**

Attack-vector templates by attempt volume and derived status (top of the graph by #attempts). Eleven canonical templates A–K absorb the bulk of effort.

| Vector | Theme (abbrev.) | Attempts | Status |
|--------|-----------------|----------|--------|
| F | format-string / attr-walk | 108 | dead |
| I | name-lookup / expr capture | 59 | **promising** |
| C | exception / frame walk | 58 | dead |
| A | builtins / introspection | 57 | dead |
| E | heap spray / type confusion | 57 | dead |
| D | sort + GC race | 56 | dead |
| G | gc / weakref | 54 | dead |
| K | misc combinations | 51 | dead |
| H | async / snapshot chain | 50 | dead |
| J | traceback / host leak | 50 | dead |
| B | filesystem / path | 42 | dead |

top of the dashboard and biases its probes accordingly. The human steers not by editing the agent's prompt mid-run but by contributing to the shared graph the agent is already required to read, a form of collaboration through a common artifact rather than through direct control. Because every attempt carries a reason and every finding carries references, the human can audit the campaign after the fact, which pure context-window or vector-store memories do not allow. The graph acts as a shared language between the human's intent and the LLM's execution.

In this deployment the human's interactions with the graph were of three concrete kinds. The first was constraint-setting before the run: recording the sandbox's known restrictions as `blocked` discoveries so the agent would not waste effort on them. The second was lead-seeding during it: adding `technique` and `cve` discoveries such as the use-after-free prior. The third was auditing after: reading the dashboard and the reason trail to write the campaign's final security report. These interactions were qualitative. The system logged the agent's reads and writes but not the human's, so we cannot report how often the analyst consulted the dashboard or how many directions a seeded lead changed. Instrumenting the human side of the loop, to measure rather than describe how a person steers an agent through a shared graph, is the most important piece of missing evidence for the UKG question, and we return to it in §7.

## 6. Deployment Experience and Data

We reconstructed the knowledge graph offline from the 679 attempt records preserved in the repository, replaying the system's own import logic into an in-memory database with the production schema and views, then querying the result. (The live database is ephemeral and not version-controlled; the markdown attempt logs are the durable record.) Table 5 and Table 6 report what the graph looks like over the full campaign.

**The graveyard is where the effort goes.** The 13 vectors the derivation rule retires as `dead` accumulated 592 of 679 attempts (87.2%). The sharper, counterfactual figure is the *avoidable re-probe* count: under "never re-probe," a vector is retired after its second scoreless attempt, so every attempt against it beyond the first two is avoidable. Summing over the dead vectors gives **566 avoidable re-probes, 83.4% of the campaign**. (Because dead vectors are entirely score-0, the ordering of attempts is irrelevant: any two tries trigger retirement and the remaining $N - 2$ are avoidable, so the metric is well-defined without reconstructing exact timestamps.) This is the re-probing the rule is *designed* to eliminate, not a measured reduction: the historical campaign did not enforce the rule throughout, and we ran no controlled comparison (§7). The number also exposes a coarse rule. Template F alone drew 108 probes because the *vector* key was coarse-grained, so genuinely different sub-techniques shared one graveyard

**Table 6**
Whole-campaign summary as derived by the graph.

| Quantity | Value |
|---|---|
| Attempts recorded | 679 |
| Distinct vectors (incl. malformed) | 42 |
| Canonical templates (A–K) | 11 |
|    attempts inside canonical templates | 642 |
| Malformed vector keys | 31 (37 attempts, 5%) |
| Attempts scoring $> 0$ | 1 |
| Best score reached | 1 (crash/panic) |
| Vectors auto-marked `dead` | 13 |
|    attempts inside dead vectors | 592 (87.2%) |
|    avoidable re-probes (rule) | 566 (**83.4%**) |
| Vectors `promising` | 1 |
| Discoveries imported (with refs) | 57 |

plot. A finer vector ontology would let the graph prune sub-techniques individually while keeping the template open, which motivates the future work in §8.

**One promising vector, no escape.** Exactly one attempt scored above zero (a crash, score 1), under the name-lookup template I, which the rule duly marked `promising`. This matches the campaign's substantive security conclusion, reported separately: Monty Round 2 withstood every Python-only attack we mounted, the previous round's use-after-free class is closed, and only a single non-exploitable latent issue was identified. For this paper the narrower point is that the graph's rule identifies the one vector worth continued attention, the one that ever produced a signal, while steering effort away from the ten that never did.

**The campaign in time.** The 679 attempts span three days: 500 on the first, 145 on the second, 34 on the third (Table 6 aggregates them). The front-loading is the swarm-style predecessor generation exhausting the obvious templates; the later, smaller batches are the research-driven KG era, where reading the graveyard before acting deliberately slows the probe rate in exchange for not repeating dead work. The snapshot kinds the sandbox surfaced are dominated by `name_lookup_snapshot` (120 occurrences), followed by `function_snapshot` (24), with `future_snapshot` (3) and `keys_snapshot` (1) rare. This is consistent with name resolution being the most productive boundary to probe, and with template I (name-lookup) being the lone promising vector.

**LLM-authored keys are noisy.** Node identity is the graph's weak point. The 11 canonical templates account for 642 of 679 attempts, but the remaining 37 attempts produced *31 distinct* malformed keys: instead of a clean template letter, the model occasionally wrote an entire prompt scaffold ("`[letter]`, REASON: . . . STRATEGY: . . .") or a fragment of its own reasoning into the key field. The asymmetry is notable: malformed keys are a *majority* of distinct nodes (31 of 42) but a small *minority* of attempts (5%). A free-text key derived from model output thus inflates the graph's node count with near-singletons while barely affecting its mass. The lesson for LLMs as KG contributors is concrete: a key taken from model output will drift, and the graph needs either a constrained vocabulary at the write boundary or a normalisation step to keep node identity stable.

**Data and artifact availability.** The figures in this section are reproducible. The knowledge-graph schema and the MCP tool definitions are part of the agent's source; the offline reconstruction script (which replays the system's own import logic over the preserved attempt logs into an in-memory database with the production schema and views) and the resulting aggregate statistics are released with this paper. The raw attempt logs contain exploit code against a live target and are not redistributed

verbatim; we provide the anonymised per-vector summary table from which every number here is computed, and the reconstruction script runs against the original logs for anyone with repository access. A single command regenerates the statistics, so the 566/83.4% and 592/87.2% figures can be checked independently.

## 7. Lessons Learned and Limitations

**What worked.**  Three properties were the most valuable. (1) *Derived status* eliminated a class of bookkeeping bugs: there was no "mark this dead" step to forget, because deadness is a query. (2) *Mandatory reasons* made the graph auditable at no extra cost and gave the derivation rule a sensible key to group on. (3) The *SELECT-only query tool* let both the human and the LLM interrogate the graph in ways the fixed views did not anticipate, without any risk to its integrity.

**Where the "graph" is thin.**  We are explicit that this is a relational knowledge base with graph-shaped semantics, not a semantic KG [5]. There is no ontology, no typed edge vocabulary, and no reasoning beyond a single CASE expression. Relations exist (attempt → vector, discovery → reference, discovery → attempt) but are implicit in foreign-key-like conventions rather than declared. For the agent's needs this was sufficient; for cross-campaign reuse, interoperability, or richer inference it is the main constraint.

**The derivation rule is crude.**  `MAX(score)=0 AND COUNT(*)>=2` is a coarse rule. Two scoreless attempts retire a vector regardless of how different they were, and a single coarse key can hide many distinct sub-techniques (the template-F problem above). The rule also has no notion of confidence or decay: a vector marked dead stays dead even if the environment changes.

**LLM-authored content needs guarding.**  The noisy-key finding generalises: any system that lets an LLM contribute node identifiers or edge types must either constrain the vocabulary at the boundary or curate afterwards. We did the former for *scores* (computed, not model-supplied) and benefited; we did not for *vector keys* and paid for it.

**Why SQLite, not a triplestore.**  We chose a relational engine deliberately, and the trade-off is instructive for the UKG audience. A triplestore or property-graph database would have provided a real edge vocabulary, path queries, and SHACL-style validation, the machinery our "thin graph" lacks. SQLite instead provided zero operational weight (a single file, no server, present in the standard library) and, for this design most importantly, *derived views as a native primitive*. The `vector_status` classification at the centre of the system is a one-line CASE expression in a view; expressing "a vector is dead when it has two scoreless attempts" as continuously-maintained, always-consistent derived knowledge was straightforward relationally and would have required either materialised inference or application code in a triple store. For a graph whose value is a *computed status* rather than a large set of asserted edges, the relational choice was right; the moment we want typed edges and cross-target inference (§8), it becomes the wrong one.

**Threats to the experience's validity.**  Our quantitative claims come from an offline reconstruction of a single agent's campaign against a single target; the 566/83.4% and 592/87.2% figures are descriptive of this deployment, not a controlled measurement of speed-up. We did not run an A/B comparison against the pre-KG (bandit) version, so we report what the graph *would* prune, not a measured reduction in wall-clock or attempts. We flag this as the most important piece of missing evidence.

## 8. Future Work

The natural next step is to make the graph genuinely semantic. A small ontology of attack techniques, with typed relations (*combines*, *requires*, *blocked-by*) and a controlled vocabulary for vector identity, would fix both the noisy-key problem and the coarse-pruning problem, letting the system retire a sub-technique without closing its parent template. Modelling *provenance* and *confidence* as first-class, rather than as a free-text `refs` blob, would support trust propagation from sources to findings. Cross-campaign reuse, carrying a graph of "what is blocked in Monty" from one bounty round to the next or across targets, would turn episodic memory into durable knowledge. Finally, the human side deserves a real interface: today the analyst reads JSON dashboards, whereas an interactive graph visualization for exploration and validation, which is squarely the UKG theme, would let a human steer the agent's attention more directly. A controlled study comparing the KG-driven loop against the prior bandit selector would turn this experience report into a measurement.

## 9. Generalising Beyond Security

Although our deployment is a security agent, neither contribution is specific to security. The pattern is general: an agent runs many *trials* against *options*, each trial yields a *measured outcome*, and the agent should not waste effort on options that the outcomes have ruled out. Any task with that shape, such as automated data cleaning (which transformation rules have failed on this column?), hypothesis search, automated debugging (which fixes have been tried and rejected?), or web automation (which navigation paths dead-end?), can use the same three ingredients: a logged trial relation, a derived view that classifies options from their outcomes, and a read-before-act discipline enforced at the tool boundary. The security specifics (a sandbox, a 0–5 escape rubric) are one instantiation of "option," "trial," and "measured outcome." What transfers is the claim that for an exploratory LLM agent, the highest-value knowledge is *negative and derived*, and that a small graph which makes ruled-out a first-class, queryable status is a better home for it than either a context window or an embedding index.

## 10. Conclusion

We reported on giving an autonomous LLM security agent a shared memory in the form of a lightweight, self-pruning knowledge graph. Two ideas carried the design: knowledge that is *derived* rather than asserted, so the most-used fact (is this worth trying?) is always a fresh query and never stale state; and contributions that are *provenance-bearing*, so the graph doubles as a human-auditable trail. On a real 679-attempt campaign, the graph's derivation rule retired 13 vectors and exposed 566 avoidable re-probes, 83.4% of all attempts, while identifying the single vector that ever produced a signal. The same data exposed the limits of a thin relational graph and of letting an LLM author node identity. As LLMs become routine users of knowledge graphs, querying them to plan and contributing to them as they act, we think small, write-guarded, derivation-driven graphs like this one are a useful and underexplored point in the design space of Human-LLM-KG collaboration.

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
