# OpenReview forum: "Vector Graveyards: A Self-Pruning Knowledge Graph as Shared Memory for a Human–LLM Security Agent"
_SEMANTiCS.cc/2026/Workshop/UKG — SEMANTiCS 2026 Workshop UKG Submission_

### Official Review · ~Fatemeh_Bagheri2 · 2026-07-11
**Accept with Major Revision**

**Rating:** 8
**Confidence:** 4

**Review:**

This paper presents an experience report on Hack Monty, an autonomous security-assessment agent redesigned to utilize a lightweight SQLite database as a shared memory substrate between an LLM orchestrator, curating sub-agents, and a human analyst. The core design relies on two principles: derived knowledge and provenance-bearing writes. Based on an offline reconstruction of 679 historical exploit attempts, the author calculates that the proposed pruning rule could potentially eliminate up to 83.4% of redundant, avoidable re-probes.
While the concept of leveraging structured relational memory to mitigate LLM amnesia and provide human-auditable logs is highly practical and interesting, the manuscript exhibits several methodology and evaluation flaws that must be addressed before publication.

•	The author claims that the system can cut down on wasted work by 83.4%, but this number is purely theoretical and calculated after the experiment was already over. The "never re-probe" rule wasn't actually running live during the campaign. Without a proper side-by-side test between the old method and this new graph system, we can't actually prove if it saves real time, cuts down on API token costs, or genuinely improves the agent's success rate.
•	Given that this manuscript is submitted to the Workshop on Users and Knowledge Graphs (UKG), the lack of empirical data regarding the human user is a critical gap. The system logs machine actions extensively, but the interactions of the human analyst, such as how frequently they consulted the dashboard, how they evaluated the reason trails, or the exact impact of human-seeded leads, remain purely descriptive and qualitative. The human element of the loop needs to be systematically observed or quantified.
•	Labeling an entire attack vector as "dead" permanently after just two scoreless attempts creates a high risk of false positives, prematurely eliminating valid paths. The system lacks a necessary status decay or recovery mechanism to re-evaluate dead paths if conditions change.
•	Relying on free-text model output to derive primary keys undermines data integrity. The author needs to discuss or implement a constrained vocabulary or a strict normalization step at the write boundary to ensure stability.

The topic of this paper is relevant to the workshop's theme, and if the author addresses the mentioned flaws in a revised version, this manuscript will make a suitable contribution to UKG.

---

### Official Review · ~Faezeh_Ensan2 · 2026-07-15
**Very hard to follow**

**Rating:** 4
**Confidence:** 3

**Review:**

The paper presents an approach for building a knowledge graph to structure the memory of an LLM-based agent. The proposed approach is implemented and evaluated in the context of an autonomous security-assessment agent. While the problem is potentially interesting, I found the paper difficult to follow, primarily because many of its central concepts are introduced without sufficient definition or background.
For example:

“Its nodes are attempts, derived attack vectors, and discoveries;
its edges are the relations between them and the references that ground discoveries in external sources.”

It is not clear what is meant by “derived attack vectors” or “discoveries.” The paper should provide sufficient background and concrete examples explaining what each node and edge type represents.

Similar terminology problems occur throughout the paper. For example, the authors write:

“a future semantic version would make attack
techniques, provenance, and confidence first-class”

It is unclear what “first-class” means in this context.

The following description is also difficult to interpret:

“The graph therefore prunes the search space rather than only storing it; we
call the set of auto-retired vectors the vector graveyard”

It is not sufficiently clear what a “vector” refers to here. Is it an attack technique, a candidate action, a hypothesis, or an embedding vector?

Another particularly unclear statement is:

“The KG is write-guarded in a way
that suits an LLM author.”

This sentence is very difficult to understand. The paper needs to explain what “write-guarded” means.

The paper’s main processing loop is also hard to understand. The role of the orchestrator is not clearly defined. Is the orchestrator itself an LLM agent? Similarly, the phrase “runs a candidate exploit
with a stated reason,” requires additional explanation. It is not clear who produces the stated reason, how the candidate exploit is selected, or how the result affects the graph.

The findings and claimed lessons are not presented clearly. The authors state:

“We draw both positive and negative lessons for
anyone building Human-LLM-KG collaboration loops.”

However, the paper does not clearly identify which findings constitute the positive lessons and which constitute the negative lessons.

Table 1 is not adequately explained and is consequently very difficult to understand. For example, the row or criterion labelled “Staleness of status” is not defined. It is also unclear why “Derived knowledge (Ours)” is marked as having no staleness of status. Derived knowledge can presumably become outdated when new evidence is obtained, the environment changes, or an earlier inference is shown to be incorrect.

I am also uncertain about the generalizability of the lessons learned section. Many of the observations appear closely tied to the specific security-assessment application.

Finally, the authors themselves acknowledge that the graph is “thin.” Technically, the proposed representation does not appear to be a semantic graph. It has no clear ontology, formal types, semantic constraints. This raises a fundamental question about why the representation is presented as a knowledge graph rather than simply a graph-structured memory.